# Single-Level Anterior Cervical Discectomy and Interbody Fusion: A Comparison between Porous Tantalum and Polyetheretherketone Cages

**DOI:** 10.3390/jpm12060986

**Published:** 2022-06-17

**Authors:** Edoardo Mazzucchi, Giuseppe La Rocca, Andrea Perna, Fabrizio Pignotti, Gianluca Galieri, Vincenzo De Santis, Pierluigi Rinaldi, Francesco Ciro Tamburrelli, Giovanni Sabatino

**Affiliations:** 1Department of Neurosurgery, Mater Olbia Hospital, 07026 Olbia, Italy; edoardo.mazzucchi@materolbia.com (E.M.); fabrizio.pignotti@materolbia.com (F.P.); gianluca.galieri@gmail.com (G.G.); giovanni.sabatino@materolbia.com (G.S.); 2Institute of Neurosurgery, IRCCS Fondazione Policlinico Universitario Agostino Gemelli, Catholic University, 00168 Rome, Italy; 3Department of Orthopedics, Mater Olbia Hospital, 07026 Olbia, Italy; perna.andrea90@gmail.com (A.P.); vincenzo.desantis@materolbia.com (V.D.S.); 4Institute of Orthopedics, IRCCS Fondazione Policlinico Universitario Agostino Gemelli, Catholic University, 00168 Rome, Italy; francescociro.tamburelli@policlinicogemelli.it; 5Unit of Radiology, Mater Olbia Hospital, 07026 Olbia, Italy; pierluigi.rinaldi@materolbia.com; 6Unit of Spine Surgery, IRCCS Fondazione Policlinico Universitario Agostino Gemelli, Catholic University, 00168 Rome, Italy

**Keywords:** cervical disc herniation, PEEK cage, stand-alone cage, subsidence, tantalum, lordosis, discectomy

## Abstract

Background: Anterior cervical discectomy and interbody fusion (ACDF) may be performed with different kinds of stand-alone cages. Tantalum and polyetheretherketone (PEEK) are two of the most commonly used materials in this procedure. Few comparisons between different stand-alone implants for ACDF have been reported in the literature. Methods: We performed a comparison between patients who underwent ACDF with either a porous tantalum or a PEEK stand-alone cage, in two spine surgery units for single-level disc herniation. Clinical outcome [Neck Disability Index (NDI), Visual Analog Scale (VAS) for pain, Short Form-36 (SF-36)] and radiological outcome (lordosis, fusion and subsidence) were measured before surgery and at least one year after surgery in both groups. Results: Thirty-eight patients underwent ACDF with a porous tantalum cage, and thirty-one with a PEEK cage. The improvement of NDI and SF-36 was significantly superior in the PEEK group (*p* = 0.002 and *p* = 0.049 respectively). Moreover, the variation of the Cobb angle for the cervical spine was significantly higher in the PEEK group (*p* < 0.001). Conclusions: In a retrospective analysis of two groups of patients with at least one year of follow-up, a stand-alone PEEK cage showed superior clinical results, with improved cervical lordosis, compared to a stand-alone porous tantalum cage. Further studies are needed to confirm these data.

## 1. Introduction

Anterior cervical discectomy and fusion (ACDF) is the surgical procedure commonly performed for the treatment of degenerative cervical disc disease [1,2]. The standard procedure implies autologous bone graft from iliac crest and anterior plating. The evidence of donor site complications has promoted the use of allograft and of osteoconductive materials instead of autologous bone graft. Porous tantalum has proved to be a metal with advantageous biomechanical and osteoconductive properties [3,4,5]; moreover, it produces slight artifact in MRI [6]. The use of porous tantalum cage in ACDF has been associated with good outcome and low rate of device-related complications at long-term follow-up [7,8].

Polyetheretherketone (PEEK) is a high-performance semi-crystalline thermoplastic polymer, whose use has been authorized in orthopedics and spine implants [9]. It has excellent mechanical properties that support its potential application as an implant, but its inertness prevents osseointegration [10]. In order to stimulate osseointegration, the implants are generally associated with autologous bone or allograft.

Anterior plating has been associated with short and long term post-operative dysphagia, probably due to the volume of the plate which irritates the esophagus. In order to reduce this complication, stand-alone cervical cages are widely used.

Many types of cages for ACDF are commercially available at the moment. A direct comparison between different kinds of cages for single-level ACDF would, in our opinion, be useful to help the surgeon choose the best implant. In the present paper, we perform a comparison of clinical and radiological outcomes between cases of ACDF with implantation of porous tantalum and those of PEEK screw-anchored cage without plating.

## 2. Materials and Methods

### 2.1. Patient Selection

We performed a retrospective comparison between consecutive patients who underwent ACDF for single-level cervical disc herniation accompanied by radiculopathy or myelopathy, with implant of two different cages: a porous tantalum cage (Trabecular Metal TM-S Cervical Fusion Device, Zimmer Spine, Minneapolis, MN, USA) or a stand-alone PEEK cage (CoRoent Small Interlock, NuVasive, San Diego, CA, USA). The interventions were carried out in two spine surgery centers: Unit of Spine Surgery, Catholic University, Rome; and the Department of Neurosurgery, Mater Olbia Hospital, Olbia.

Patients with a history of previous surgical intervention on cervical spine or with a follow-up duration of less than one year were excluded.

### 2.2. Operative Techniques

A standard anterior Smith-Robertson approach was performed in both groups. A Caspar distractor is used to obtain gentle distraction of the vertebral bodies. With the aid of an operative microscope, the vertebral plates were prepared with a high-speed drill or with curettes, after discectomy. The posterior longitudinal ligament was opened, and optimal neural decompression was confirmed.

The PEEK cage was packed with a biphasic calcium-phosphate bone graft substitute (AttraXPutty, NuVasive Inc., San Diego, CA, USA) and three screws with a 40° angulation were inserted to anchor the cage (two in the superior vertebral body and one in the inferior one). In the tantalum group, the cage was inserted without any adjunctive bone graft (Figure 1 and Figure 2). Both cages have a 7° lordotizing angulation.

### 2.3. Post-Operative Management

All the patients were ambulating the day after surgery. For the patients in the tantalum group, a cervical collar (Aspen Vista^®^) was prescribed for the first month after surgery. No collar was prescribed for the PEEK group. Physical therapy was recommended on a case-by-case basis.

### 2.4. Clinical and Radiological Data

Patients underwent preoperative MRI and cervical spine X-ray (anteroposterior and neutral lateral projections). A cervical spine X-ray was performed after surgery and at last follow-up. Patients were evaluated pre-operatively and at follow-up with Visual Analogic Scale (VAS) for pain, Neck Disability Index (NDI) and Short Form-36 (SF-36).

The presence of subsidence was investigated in follow-up imaging. It was defined as a reduction of more than 2 mm of disc height or cage migration (see [11]).

The Cobb angle was measured for global cervical spine (Cobb-c) and for the operated segment (Cobb-s) [12] pre-operatively and at follow-up to investigate the effect of the procedure on lordosis.

Moreover, fusion at follow-up was evaluated according to the Bridwell classification [13].

### 2.5. Statistical Analysis

Statistical analyses were performed using GraphPad Prism 5.01 (GraphPad Software Inc., San Diego, CA, USA). Radiological and clinical data were expressed as means ± standard deviation. The data were analyzed using the Mann-Withney U test or the Chi-squared test as appropriate.

## 3. Results

### 3.1. Demographic and Operative Data

The total number of patients included in the study was 69, 38 of whom were in the tantalum group and 31 in the PEEK group. Demographic data are summarized in Table 1. Patients in the PEEK group were significantly older (41 ± 6.3 vs. 50 ± 10.5) than patients in the tantalum group (*p* < 0.001). The operative time was significantly different between the two groups (*p* < 0.001).

### 3.2. Clinical Outcome

Clinical parameters improved at follow-up in both groups (Table 2). On comparing the two groups, NDI and SF-36 were significantly better at follow-up (*p* = 0.04 and *p* < 0.001 respectively) in the PEEK group and significantly more improved (*p* = 0.002 and *p* = 0.049 respectively). Pre-operative VAS was significantly higher in the PEEK group (*p* = 0.014), although we have not found any significant difference between the two groups at follow-up, or on analyzing its variation from pre-operative to follow-up (Table 2).

### 3.3. Radiographic Outcome

The variation of the Cobb angle for the cervical spine was significantly higher in the PEEK group (*p* < 0.001). Segmental Cobb angle was significantly different before surgery (*p* < 0.001) and at follow-up (*p* < 0.001), as was the variation (*p* < 0.001) between these two time points under consideration. Subsidence and fusion were comparable in the two groups (Table 3).

## 4. Discussion

ACDF is a complex procedure in which fusion had been classically achieved with autologous bone graft (from iliac crest) and plating. Donor site complication (especially pain) [14] promoted the use of cervical cages of various materials; moreover, the evidence of a higher rate of post-operative dysphagia with anterior plate [15] progressively led to the withdrawal of anterior plating for single level cervical disc herniation. Comparative studies showed lower rate of dysphagia with higher risk of subsidence without plating. Nevertheless, subsidence has not been associated with worse clinical outcome in a systematic review [16].

The choice of the kind of cervical cage used in ACDF depends on a number of factors. We found scarce literature performing comparisons between different kinds of stand-alone cages [12,17,18,19,20]. In the present study, we compared two groups of patients who underwent surgery in two spine surgery centers for cervical disc herniation with implantation either of a porous tantalum cage or of a PEEK cage. NDI and SF-36 at follow-up were both significantly better in the PEEK group, with a more significant improvement (pre-operative—follow-up). Moreover, radiological data showed a significant difference when considering variation of lordosis of cervical spine and of Cobb-s.

The presented data suggests that the implanted PEEK cage provides good clinical performance compared with a tantalum implant that has already showed significant clinical benefits in the previous literature [8]. PEEK cages have already been compared with titanium cages by Niu et al. [19] who reported superior radiological outcome in the PEEK group, concerning interspace height and radiographic fusion. They related the superiority of the outcome directly to the cage material: PEEK has demonstrated absence of cytotoxicity and mutagenicity [21]; is biocompatible, nonabsorbable, corrosion resistant [22]; and, most importantly, has a modulus of elasticity similar to the bone. This feature has been related to a lower risk of subsidence. In a comparison between titanium and PEEK cage, Cabraja et al. [20] did not find significant clinical or radiological differences. They observe that the material of the cage is only one of the multiple factors that lead to a different outcome. Indeed, the design, shape, size and surface architecture of the cage are also relevant. Moreover, the surgical technique may influence the radiological and clinical outcome, for example through endplate preparation and use of distraction during surgery. The specific PEEK cage utilized is associated with the placement of three screws. This additional implant should lower the risk of subsidence. The adjunctive phase of surgery necessary to screw did not significantly prolong the operative time (which is even shorter for the PEEK group, in this small case series). Furthermore, it allows immediate mobilization without collar. On the other hand, we observed some cases of radiological subsidence, with neither significant mobilization of the cage, nor correlation with neurological impairment.

The rate of radiological fusion appears optimal at follow-up. The lack of osseointegration of this inert material is apparently overcome by the use of the bone substitute and screws that immediately reduces the movements.

The described PEEK cage provides several advantages: favorable biomechanics of the material, immediate “internal immobilization” with screws, and the use of a bone substitute that avoids the need for harvesting autologous bone. All these features could contribute to the improved outcome.

The small sample size does not make it possible to draw strong conclusions. Nevertheless, the PEEK group shows good clinical outcome which could be related with improved post-operative lordosis, as suggested by the previous literature [23]. Moreover, the immediate mobilization without cervical collar may be an adjunct positive element allowing a return to physiological muscular activity and resulting in a faster recovery after surgery [24].

### Limitations

The retrospective nature of this investigation is the principal limitation of the study, as it does not allow having two homogeneous groups. Indeed, it must be noted that the two groups are significantly different before surgery, both in terms of age and of Cobb-s. Results must therefore be interpreted taking into account this difference which is a consequence of the retrospective design of the study and of the relatively small study size. Moreover, a longer follow-up duration would have made it possible to evaluate how the two implants are integrated in the cervical spine biomechanics, thus also allowing, for example, an evaluation of the incidence of adjacent segment degeneration. This will be the object of future studies.

Furthermore, we have not considered numerous variables that should be considered in the choice of a specific cage. Relevant issues to be considered notably include: the availability of the implant, the preference of the surgeon, the cost of the implant, and the ergonomics of the surgical instruments designed for each cage.

## 5. Conclusions

A PEEK anchored cage for ACDF has showed some advantages over a tantalum cage at least one year after surgery when considering lordosis, disability and physical health indices, in a retrospective analysis of a relatively small population of patients. Future studies are needed to evaluate long term effects on a larger population of patients.

## Figures and Tables

**Figure 1 jpm-12-00986-f001:**
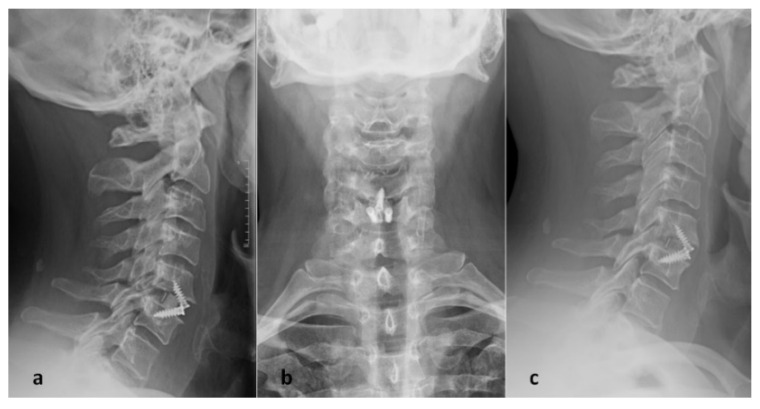
Cervical X-ray immediately after surgery (**a**,**b**) and one year after surgery (**c**) in a patient who underwent ACDF with PEEK cage (CoRoent Small Interlock, NuVasive, San Diego, CA, USA) implantation at C5–C6 level.

**Figure 2 jpm-12-00986-f002:**
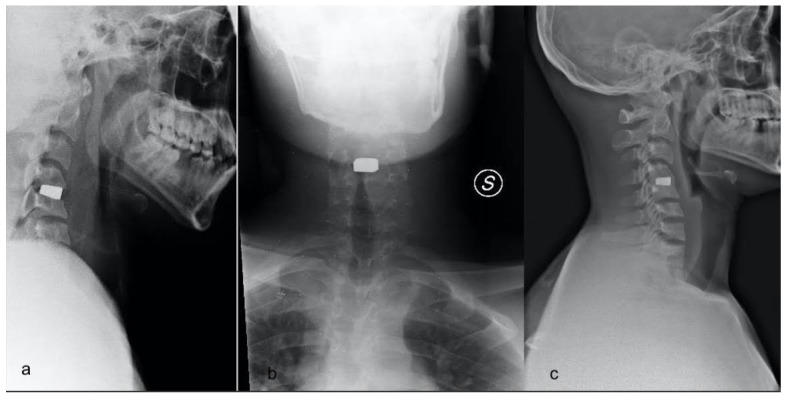
Cervical X-ray immediately after surgery (**a**,**b**) and one year after surgery (**c**) in a patient who underwent ACDF with tantalum cage (Trabecular Metal TM-S Cervical Fusion Device, Zimmer Spine, Minneapolis, MN, USA) implantation at C4–C5 level.

**Table 1 jpm-12-00986-t001:** Demographic and clinical data.

Variable	Tantalum	PEEK	*p*-Value
N° of patients	38	31	-
Sex (male:female)	15:23	14:17	0.634
Age (years)	41 ± 6.3	50 ± 10.5	<0.001
Smoke (yes:no)	16:22	14:17	0.101
Follow-up (days)	616.7 ± 93	544.3 ± 114.5	0.016
Operative time	100.4 ± 25.2	78.9 ± 19.2	<0.001
Length of stay	2.2 ± 0.4	2.2 ± 0.5	0.588
Location	-	-	0.101
C3C4	0	2	-
C4C5	7	5	-
C5C6	17	20	-
C6C7	14	3	-
C7D1	0	1	-

PEEK: polyetheretherketone.

**Table 2 jpm-12-00986-t002:** Clinical outcome.

	Tantalum	PEEK	*p*-Value
**NDI**			
pre-operative	24.6 ± 5.3	27.9 ± 9.9	0.09
at follow-up	12.7 ± 4.4	9.6 ± 8.8	0.04
pre-FU	11.8 ± 5.9	18.4 ± 9.9	0.002
**VAS**			
pre-operative	6.9 ± 1.4	7.8 ± 1.7	0.014
at follow-up	2.4 ± 1.1	2.6 ± 2	0.926
pre-FU	4.5 ± 1.6	5.2 ± 2.2	0.193
**SF-36**			
pre-operative	37.3 ± 6.8	43 ± 19.3	0.411
at FU	48.5 ± 5.3	64.7 ± 20.7	<0.001
pre-FU	−11.1 ± 7.1	−21.7 ± 19.2	0.049

NDI: Neck Disability Index; VAS: Visual Analogic Scale; SF-36: Short Form-36; pre-FU: variation between pre-operative and follow-up.

**Table 3 jpm-12-00986-t003:** Radiological outcome.

	Tantalum	PEEK	*p*-Value
**Cobb-c** (degree)			
pre-operative	7.9 ± 2.1	8 ± 6.7	0.08
at follow-up	9.8 ± 2.5	11.9 ± 7.5	0.918
pre-FU	−1.8 ± 1.4	−4 ± 1.9	<0.001
**Cobb-s** (degree)			
pre-operative	0.6 ± 1	2.5 ± 2.8	<0.001
at follow-up	2.9 ± 0.9	6.9 ± 2.9	<0.001
pre-FU	−2.2 ± 0.8	−4.3 ± 2.2	<0.001
**Subsidence** (Number of patients)	5	7	0.304
**Fusion-Bridwell classification** (Number of patients)	-	-	0.018
I	19	26	-
II	13	5	-
III	5	0	-
IV	1	0	-

Cobb-c: angulation of C2–C7 tract; Cobb-s: angulation of the operated segment; pre-FU: variation between pre-operative and follow-up.

## Data Availability

The data presented in this study are available on request from the corresponding author.

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
