# Peer review of "Single-Level Anterior Cervical Discectomy and Interbody Fusion: A Comparison between Porous Tantalum and Polyetheretherketone Cages"

_jpm, 2022, doi:10.3390/jpm12060986_

Round 1
Reviewer 1 Report
First of all thank you very much for the work of the author. The current study focused on the comparsion of between patients who underwent ACDF with a porous tantalum or a PEEK stand-alone cage. The quality of the manuscript was acceptable. But some areas need to be improved. As an important postoperative complication, incidences of dysphagia wasn’t recorded in both 2 groups. In the part of discussion, there is too little interpretation of the difference between tantalum or a PEEK. There is a sentence“We found scarce literature performing comparison between different kinds of stand-alone cages.” However, I found no fewer studies of this type after searching. These studies should be cited to make complementary interpretation. Overall, I hope the author will further improve this manuscript.
Author Response
First of all thank you very much for the work of the author. The current study focused on the comparsion of between patients who underwent ACDF with a porous tantalum or a PEEK stand-alone cage. The quality of the manuscript was acceptable. But some areas need to be improved. As an important postoperative complication, incidences of dysphagia wasn’t recorded in both 2 groups.
- We would like to thank the reviewer for the appreciation of the work. We do not perform ENT evaluation or fibrolaryngoscopy routinely. Mild swallowing discomfort in the immediate post operative period was considered not significant. One month after surgery, no patient reported dysphagia, probably as a consequence of the relatively small sample size.
In the part of discussion, there is too little interpretation of the difference between tantalum or a PEEK.
- The discussion has been modified
There is a sentence“We found scarce literature performing comparison between different kinds of stand-alone cages.” However, I found no fewer studies of this type after searching. These studies should be cited to make complementary interpretation.
- We thank the reviewer for the suggestion. We added relevant references.
Overall, I hope the author will further improve this manuscript.
We would like to thank the reviewer for this possibility to improve this article.
Reviewer 2 Report
General impression
In this article, the authors performed a comparison of clinical and radiographical outcome of anterior cervical discectomy and fusion (ACDF) with implantation of porous tantalum of polyetheretherketone (PEEK) screw-anchored cage without plating. And they conducted the results that a stand-alone PEEK cage showed superior clinical results, with improved cervical lordosis, compared with a stand-alone porous tantalum cage. I believe the information in this study must be valuable for the physicians to manage cervical spine diseases.
The methodology of this study was precisely explained and acceptable. Also, the limitations of this project were well indicated. For these reasons, I think this manuscript is appropriate for publication.
However, I have a couple of minor requests to be revised as stated below. After they have been resolved, I will judge this manuscript can be accepted and published by JPM.
1. Page 4 line 1
“standalone” should be replaced by “stand-alone” to coordinate with other places.
2. Figure 1 and Figure 2
In the paragraph of the Operative techniques in the materials and methods section, the explanations of surgical procedures are firstly Tantalum followed by PEEK. However, Figure 1 is the case of PEEK and Figure 2 is the case of Tantalum. I think the case presentations of Figure 1 and Figure 2 may be converted to facilitate the understandings of readers.
3. Table III
I recommend the explanations of Cobb-c and Cobb-s should be added just like as NDI, VAS, and so on in the Table II because they must be helpful for readers to understand this table.
Author Response
In this article, the authors performed a comparison of clinical and radiographical outcome of anterior cervical discectomy and fusion (ACDF) with implantation of porous tantalum of polyetheretherketone (PEEK) screw-anchored cage without plating. And they conducted the results that a stand-alone PEEK cage showed superior clinical results, with improved cervical lordosis, compared with a stand-alone porous tantalum cage. I believe the information in this study must be valuable for the physicians to manage cervical spine diseases.
The methodology of this study was precisely explained and acceptable. Also, the limitations of this project were well indicated. For these reasons, I think this manuscript is appropriate for publication.
- We would like to thank the Reviewer for the appreciation of the manuscript.
However, I have a couple of minor requests to be revised as stated below. After they have been resolved, I will judge this manuscript can be accepted and published by JPM.
- Page 4 line 1
“standalone” should be replaced by “stand-alone” to coordinate with other places.
- It has been modified
- Figure 1 and Figure 2
In the paragraph of the Operative techniques in the materials and methods section, the explanations of surgical procedures are firstly Tantalum followed by PEEK. However, Figure 1 is the case of PEEK and Figure 2 is the case of Tantalum. I think the case presentations of Figure 1 and Figure 2 may be converted to facilitate the understandings of readers.
- It has been modified
- Table III
I recommend the explanations of Cobb-c and Cobb-s should be added just like as NDI, VAS, and so on in the Table II because they must be helpful for readers to understand this table.
- The table has been modified